# KNOWLEDGE-TO-VERIFICATION: UNLOCKING REINFORCEMENT LEARNING WITH VERIFIABLE REWARDS FOR LLMS IN KNOWLEDGE-INTENSIVE DOMAINS

## ABSTRACT

Reinforcement learning with verifiable rewards (RLVR) has demonstrated promising potential to enhance the reasoning capabilities of large language models in domains such as mathematics and coding. However, its application has not been effectively extended to knowledge-intensive domains due to the issue of unverifiable answers and the Limited of high-quality verifiable data. Furthermore, existing RLVR methods suffer from two inherent drawbacks. First, they focus solely on the final answer's correctness while ignoring the verification of reasoning process, which can lead to flawed reasoning. Second, this reliance on a final answer creates a sparse binary reward signal that destabilizes the training process. To address these challenges, we propose **K**nowledge-**t**o-**V**erification (**K2V**), a framework that extends RLVR to knowledge-intensive domains and enabling verification of the model's reasoning process, without any human supervision. K2V is built on two key observations. First, structured knowledge is easier to verify than unstructured knowledge. Second, a complex reasoning process can be decomposed into a series of verifiable sub-tasks. Specifically, K2V first constructs a knowledge graph from text and then models the knowledge verification as a text-based knowledge graph completion task, thereby automatically synthesizing large-scale verifiable question-answering (QA) pairs. Then, K2V generates a checklist of sub-tasks for each QA pair. The model's reasoning process is verified by evaluating these sub-tasks, which in turn provides dense rewards. Extensive experiments demonstrate that K2V enhances the model's fundamental reasoning skills, which improves its reasoning capabilities in knowledge intensive domains while keeping its capabilities in general domains stable, or even slightly improved. Our work suggests that extending RLVR to knowledge-intensive domains through automated data synthesis is a promising direction. Meanwhile, verifying the reasoning process proves to be an effective method for overcoming the inherent drawbacks of RLVR. The code is available at `https://anonymous.4open.science/r/k2v-C123`.

## 1 INTRODUCTION

Recent large language models (LLMs), such as OpenAI-o1 (Jaech et al., 2024), DeepSeek-R1 (Guo et al., 2025a), and Qwen3 (Yang et al., 2025a), have demonstrated remarkable progress in reasoning. Central to this progress is reinforcement learning with verifiable rewards (RLVR) (Shen et al., 2025; Peng et al., 2025; Stojanovski et al., 2025), which drives the model to self-explore during training by comparing its outputs against a verifiable ground truth, thereby enhancing its capacity for complex problem-solving.

However, current RLVR methods are confined to mathematical reasoning (Hu et al., 2025; Zeng et al., 2025; Liu et al., 2025) and coding tasks (He et al., 2025; Luo et al., 2025; Cui et al., 2025), lack the transferability to knowledge-intensive domains (*e.g.* agriculture, medicine, *etc.*). This narrow focus can be attributed to two main reasons (1) Unverifiable Answers: In mathematics, the correctness of a model's response to certain problems, such as computation or multiple-choice questions, can be directly validated by a rule-based verifier. (Hu et al., 2025). As for the domain of code, unit tests can be directly executed on model-generated code within a sandboxed environment (Luo et al., 2025). However, in knowledge-intensive domains, answers are typically available only in

a free-form format. These answers exhibit high diversity and complexity, which means they cannot be directly validated by rule-based verifiers. (2) Limited Data: From a data source perspective, the internet contains large-scale verifiable math problems and executable open-source code (Ma et al., 2025). In contrast, data scraped for knowledge-intensive domains is predominantly unverifiable text or of low quality. From a data synthesis perspective, because problems in mathematics and code are easier to verify, synthesizing new verifiable problems is also more straightforward (Yang et al., 2025a). However, knowledge-intensive domains lack effective data synthesis methods and rely on costly, expert-level manual annotation (Dubois et al., 2024).

Moreover, current RLVR methods suffer from two inherent drawbacks across various domains (1) Flawed Reasoning: These methods focus solely on the correctness of the final answer, ignoring the validity of the reasoning process. This reward mechanism can lead the LLMs' reasoning to exhibit issues like linguistic incoherence (Guo et al., 2025a) and unfaithful reasoning (Chen et al., 2025a). (2) Sparse Rewards: Awarding a binary reward based only on the final answer creates an overly sparse reward signal, which increases the variance of the policy gradient estimate, introduces noise to the training, and results in unstable learning and slower convergence (Su et al., 2025).

In this paper, we investigate how to advance current RLVR methods by addressing their fundamental limitations. We formulate our research around two central questions:

- **RQ1:** How can we address the issue of unverifiable answers, thereby resolving the limited data and ultimately extending the RLVR to knowledge-intensive domains ?

- **RQ2:** How can we mitigate the issue of flawed reasoning, thereby resolving the sparse rewards and ultimately overcome the inherent drawbacks of current RLVR ?

To answer these questions, we propose **K**nowledge-**t**o-**V**erification (**K2V**), which synthesizes verifiable question-answering (QA) pairs in knowledge-intensive domains without requiring any human annotation or seed datasets (Wang et al., 2022; Yu et al., 2025a), while also enabling the verification of the LLMs' reasoning process. *The core observation of K2V is that knowledge in a structured representation is more easily verifiable than unstructured knowledge*. (*e.g.* knowledge within a dense, unstructured text becomes more easily verifiable when organized into a table). To this end, K2V begins by structuring knowledge from the corpus into a Knowledge Graph (KG). It then models knowledge verification as a text-based Knowledge Graph Completion (KGC) task (Hogan et al., 2021; Yao et al., 2025), effectively converting knowledge from large-scale unstructured texts into verifiable QA pairs. Furthermore, *we posit that directly verifying the correctness of an LLM's reasoning process is challenging. However, this task can be decomposed into multiple, binary-verifiable sub-tasks*. (*e.g.* when solving a physics problem, the reasoning process can be verified by checking sub-tasks such as whether the correct formula was chosen and whether the correct values were substituted). Specifically, we introduce a checklist-style verification method that generates a checklist for each QA pair. This checklist consists of several independent, human-interpretable, and binary-verifiable sub-tasks that describe the criteria for a correct reasoning process.

Experiments across three distinct domains demonstrate that K2V enhances reasoning capabilities in knowledge-intensive domains without compromising, and even improving, the model's general capabilities. In the agriculture domain, leveraging Qwen2.5-3B (Qwen et al., 2025) as backbone, K2V achieves 63.04 on SeedBench (Ying et al., 2025), outperforming the much larger Qwen2.5-72B-Instruct (62.06). Notably, without using any general data, K2V's general performance is on par with the official instruction-tuned model Qwen2.5-3B-Instruct, even achieving a increase of 23.99% on BBH (Suzgun et al., 2022) and 5.24% on GPQA-Diamond (Rein et al., 2024). Further experiments based on Llama3.1 (Dubey et al., 2024) and Llama3.2 (Meta, 2024) confirm the robust performance of K2V.

The contributions of this work can be summarized as below. We propose K2V, a framework that efficiently extends RLVR to knowledge-intensive domains. In K2V, we introduce two novel designs. First, we model knowledge verification as a text-based KGC task, which enables the cost-effective synthesis of verifiable QA pairs. Second, we employ a checklist-style verification method to validate the model's reasoning process, which provides a dense reward signal. Extensive experiments demonstrate that K2V enhances the model's fundamental reasoning skills, which in turn significantly improves its capabilities in knowledge-intensive domains while keeping general capabilities stable or even slightly improved.

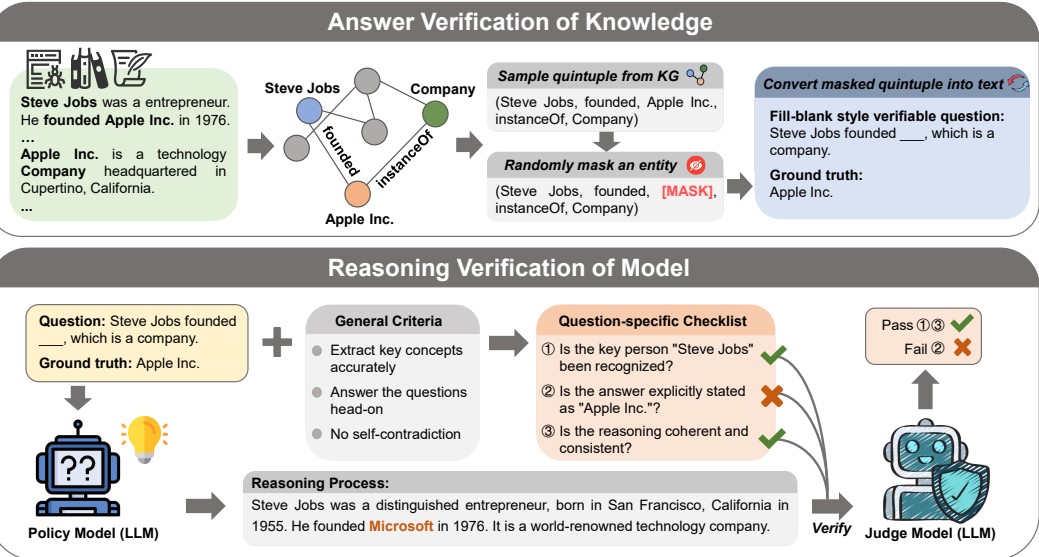

Figure 1: Overview of K2V. **(Top)** K2V begins by constructing a KG from unstructured text. It then samples quintuples from the KG and randomly masks an entity. This masked quintuple is then converted into a fill-blank style verifiable question. **(Bottom)** Given a question, Policy Model generates a reasoning process. To verify this reasoning process, K2V generates a Question-Specific Checklist by instantiating a set of General Criteria based on the given question and its ground truth. A Judge Model then verifies the reasoning process against each item in the checklist one by one.

## 2 KNOWLEDGE-TO-VERIFICATION

In this section, we first introduce how to synthesize QA pairs with verifiable answers in knowledge-intensive domains (Section 2.1). We then present checklist-style verification, which not only validates the correctness of the model's reasoning process but also provides dense rewards (Section 2.2). Finally, we discuss how to combine the two distinct rewards derived from answer verification and reasoning verification (Section 2.3). An overview of K2V is shown in Figure 1.

### 2.1 ANSWER VERIFICATION OF KNOWLEDGE

Knowledge-intensive domains have large-scale knowledge-rich, unstructured corpus that can be easily and inexpensively crawled from the internet in a scalable manner. The goal of K2V is to convert the knowledge within unstructured corpus into verifiable QA pairs. We will begin by explaining the fundamental concepts of text-based Knowledge Graph Completion (KGC), as introduced below.

**Text-based KGC.** A KG is a structured representation of factual knowledge, formally defined as $\mathcal{G} = \{\mathcal{E}, \mathcal{R}, \mathcal{T}\}$, where $\mathcal{E}$ is the set of entities, $\mathcal{R}$ is the set of relations, and $\mathcal{T} \subseteq \mathcal{E} \times \mathcal{R} \times \mathcal{E}$ represents the set of factual triples. Each triple $(h, r, t) \subseteq \mathcal{T}$ consists of a head entity $h$, a relation $r$, and a tail entity $t$. **The goal of conventional KGC is entity prediction:** Given a triple with a missing entity, formally expressed as $(h, r, ?)$ for a missing tail or $(?, r, t)$ for a missing head, the task is to predict the missing entity from the set $\mathcal{E}$ (Ji et al., 2021).

Different from the conventional KGC, **K2V reformulates the task as text-based KGC, where knowledge verification is represented as fill-blank style verifiable QA pairs.** Meanwhile, to ensure the generated questions possess sufficient context for complex reasoning, K2V operates not on individual triples, but on quintuples. A quintuple $\tau$ is defined as:

$$\tau = (e_1, r_1, e_2, r_2, e_3) \tag{1}$$

which represents two consecutive triples, $(e_1, r_1, e_2)$ and $(e_2, r_2, e_3)$, forming a path of length two in the KG. This structure provides a richer context than a single triple, making the generated questions more diverse and the reasoning task more challenging and realistic.

To generate a QA pair, we first randomly mask one of the three entities ($e_1$, $e_2$, or $e_3$) to create a masked quintuple $\tau_{\text{masked}}$, e.g., $(e_1, r_1, \texttt{[MASK]}, r_2, e_3)$. We then employ a textualization function, $\mathcal{F}_{\text{text}}$, powered by a generative LLM, to convert the masked quintuple into a fill-blank question.

$$Q_{\text{blank}} = \mathcal{F}_{\text{text}}(\tau_{\text{masked}}) \tag{2}$$

The ground truth for this fill-blank question $Q_{\text{blank}}$ is the name of the masked entity. The prompt of $\mathcal{F}_{\text{text}}$ is shown in Table 7 in Appendix.

**QA Pairs Synthesis.** Text-based KGC provides a pathway to convert quintuples into fill-blank style verifiable QA pairs. Consequently, once a KG is constructed in the knowledge-intensive domains (Chen et al., 2025b), we can synthesize numerous verifiable QA pairs at a low cost by sampling quintuples from the KG.

First, K2V constructs a KG from a diverse, unstructured corpus crawled from the internet, which includes web pages, scientific literature, and textbooks. To convert this raw text into KG, K2V leverages a generative LLM to perform Named Entity Recognition (NER) (Keraghel et al., 2024) and Relation Extraction (RE) (Zhao et al., 2024) on the chunks from the corpus (see Table 8 in Appendix for the prompt). Entities from different chunks are then linked to construct the final KG.

After the KG is constructed, K2V directly samples quintuples from the KG. For each sampled quintuple $\tau$ (see Equation 1), K2V randomly selects one entity $e_{\text{masked}} \in \{e_1, e_2, e_3\}$ to be masked, and then uses the $\mathcal{F}_{\text{text}}$ (see Equation 2) to convert the masked quintuple into a sentence. This sentence is a fill-blank style question, with the ground truth being the masked entity $e_{\text{masked}}$ .

K2V operates entirely from scratch, requiring no human annotation or seed datasets (Wang et al., 2022; Yu et al., 2025a). This approach is both efficient and scalable: the more unstructured corpus are available, the more verifiable QA pairs can be synthesized (**RQ1**).

## 2.2 REASONING VERIFICATION OF MODEL

Given an input question $x$ and a policy model $\pi_\theta$ parameterized by $\theta$, a reasoning process $z$ and a response $\hat{y}$ are sampled from the policy, denoted by $z, \hat{y} \sim \pi_\theta(\cdot|x)$. The goal of K2V is to verify the correctness of the reasoning process, thereby providing a dense reward. In particular, we first discuss how checklist-style verification decomposes a complex reasoning verification task into multiple, binary-verifiable sub-tasks, as introduced below.

**Checklist-Style Verification.** The reasoning process $z$ of a policy model is typically a lengthy, open-ended text that describes the policy's thinking details for a given question $x$. Due to the lack of evaluation criteria, directly verifying the correctness of $z$ is challenging. Motivated by this, we propose checklist-style verification. For each question $x$, we establish a checklist, which is formally represented as a set of $k$ verifiable criteria:

$$CL_x = \{c_1, c_2, \ldots, c_k\} \tag{3}$$

where each criterion $c_i$ is a binary-verifiable criterion that assesses a desirable property of a reasoning process. These criteria evaluate the policy's reasoning process from different perspectives, and collectively, they form a checklist $CL_x$ for a comprehensive assessment of reasoning quality. Most importantly, **the checklist $CL_x$ is question-specific:** a unique checklist is generated for each question $x$. This enables a tailored evaluation of reasoning quality for different questions.

To perform the verification, K2V employs a generative LLM as the judge model, denoted as $J$. The judge model assesses the reasoning process $z$ against each criterion $c_i$ in the checklist $CL_x$ one by one (see Table 10 in Appendix for the prompt). For each pair $(z, c_i)$, the judge model outputs a binary score $v_i \in \{0, 1\}$, where $v_i = 1$ indicates that $z$ satisfies the criterion $c_i$, and $v_i = 0$ otherwise. We can define this verification process as:

$$v_i = J(z, c_i) \tag{4}$$

Once the judge model evaluates all criteria, K2V aggregates these binary scores to compute a pass rate $p \in [0, 1]$, representing the proportion of criteria that the reasoning process $z$ successfully meets:

$$p = \frac{1}{k} \sum_{i=1}^{k} v_i \tag{5}$$

where $v_i$ is computed by Equation 4. This approach decomposes the intractable reasoning verification task into a series of binary-verifiable sub-tasks. Additionally, the resulting pass rate $p$ can serve as a dense reward signal (**RQ2**).

**Checklist Synthesis.** Checklist-style verification enables us to validate the model's reasoning process while providing a dense reward signal. The subsequent challenge is to synthesize a problem-specific checklist $CL_x$ (see Equation 3) for any given prompt $x$. We propose a two-stage synthesis pipeline, as introduced below.

First, we define a set of general criteria, formally denoted as:

$$G = \{g_1, g_2, \ldots, g_N\} \tag{6}$$

where each $g_i$ is a high-level, universal principle that characterizes a high-quality reasoning process, independent of any specific question. We developed a set of general criteria based on the scoring rubrics from the AP Biology Course and Exam Description [1]. An example is shown in Table 5.

Second, for a given QA pair consisting of a question $x$ and a ground truth $y$, we feed both $x$, $y$ and the set of general criteria $G$ into a generative LLM, denoted as $S$. This LLM is prompted to instantiate the abstract principles in $G$ into a concrete, question-specific checklist $CL_x$ suitable for verifying reasoning related to (see Table 9 in Appendix for the prompt). This synthesis process can be expressed as:

$$CL_x = \{c_1, \ldots, c_k\} \sim S(\cdot|x, y, G) \tag{7}$$

This automated pipeline enables K2V to generate tailored checklists for a wide variety of questions, ensuring that our verification is both scalable and contextually appropriate.

## 2.3 REWARD COMPOSITION

In the preceding sections, we introduced two distinct verification mechanisms: answer verification for validating the correctness of knowledge, and reasoning verification for assessing the quality of the reasoning process. Following DeepSeek-R1 (Guo et al., 2025a), we form the training signal by directly summing the individual reward components.

$$R_{\text{total}} = R_{\text{format}} + R_{\text{answer}} + R_{\text{reasoning}} \tag{8}$$

**Format Reward ($R_{\textbf{format}}$).** The format reward, $R_{\text{format}}$, encourages the policy model to generate outputs in a structured and parsable format. The desired output template is consistent with that of DeepSeek-R1 (Guo et al., 2025a). A maximum score of 0.75 is awarded if the output perfectly adheres to this template.

**Answer Reward ($R_{\textbf{answer}}$).** The answer reward, $R_{\text{answer}}$, provides a binary reward based on the final answer's correctness. If the model's predicted answer $\hat{y}$ exactly matches the ground truth $y$, a reward of $\alpha$ is granted; otherwise, the reward is 0. In our main experiments, we set $\alpha = 6$. We also conduct a sensitivity analysis on the value of $\alpha$ to investigate how different answer reward values affect model performance, as detailed in Section 3.4.

**Reasoning Reward ($R_{\textbf{reasoning}}$).** The reasoning reward, $R_{\text{reasoning}}$, is a dense signal derived from our checklist-style verification (Section 2.2). This reward encourages the model to produce logically coherent reasoning processes, and is set directly to the pass rate $p$ calculated in Equation 5.

## 3 EXPERIMENTS

In this section, we empirically investigate the effectiveness of K2V. Our analysis includes not only an evaluation of the overall model performance but also ablation studies of its components and a sensitivity analysis to varying reward values.

---

[1]The AP Biology Course and Exam is a program run by the U.S. College Board that provides a college-level introductory biology curriculum to high school students. The scoring rubrics for this exam can be found in: https://apcentral.collegeboard.org/media/pdf/ap-biology-course-and-exam-description.pdf

## 3.1 EXPERIMENTAL SETUPS

**Backbone Models.** We use Qwen2.5 (Qwen et al., 2025), Llama3.1 (Dubey et al., 2024), and Llama3.2 (Meta, 2024) series models as our backbone. For the Qwen series, our K2V-3B and K2V-7B models are built upon Qwen2.5-3B-Base and Qwen2.5-7B-Base, respectively, while all other baselines are implemented on Qwen2.5-3B-Base. For the Llama series, K2V-3B and K2V-8B are based on Llama-3.2-3B-Instruct and Llama-3.1-8B-Instruct, respectively.

**Training Corpus.** K2V's training begins with an unstructured text corpus. We trained separate models in the domains of agriculture, medicine and law. For agriculture, we use RiceCorpus (Yang et al., 2025b), a corpus built by crawling extensive resources from the internet, including 1.4 million papers published over the past 40 years and 1,207 related books, totaling 3,397.49 GB. This data covers multiple disciplines such as molecular biology, plant breeding, and agronomic practices. The raw corpus was processed through a meticulously designed cleaning pipeline, yielding a high-quality corpus of approximately 1.1B tokens. For medicine, we use shibing624-medical-pretrain (shibing624, 2024), a Chinese corpus sourced from online medical encyclopedias and textbooks. We randomly downsample 23.51 MB of text from this corpus for training. For law, we use DISC-Law-SFT (Yue et al., 2023), a dataset of QA pairs for supervised fine-tuning (SFT). We directly concatenate the questions and answers, then randomly downsample 20.69 MB of the text as our training corpus. This indirectly demonstrates K2V's robustness to varied corpus sources.

**Evaluation.** In addition to evaluations in agriculture, medicine, and law, we also tested the model's performance on general benchmarks. All evaluations are conducted in a zero-shot setting, and we report the pass@1 results under greedy sampling. For agriculture, to ensure an objective and fair evaluation, we evaluate on the multiple-choice subset of SeedBench (Ying et al., 2025) and conduct experiments on agriculture-related subsets of CMMLU(Li et al., 2023) and MMLU (Hendrycks et al., 2020). For medicine, we evaluate on MedQA-MCMLE (Jin et al., 2021), as well as on medicine-related subsets of CMMLU and MMLU. For law, we evaluate on LawBench (Fei et al., 2023), as well as on law-related subsets of CMMLU and MMLU. The details of subset selection for all three domains are provided in the Appendix C.3.Finally, for general benchmarks, we evaluate on BBH (Suzgun et al., 2022), MATH-500 (Hendrycks et al., 2021; Lightman et al., 2023b), GSM8K (Cobbe et al., 2021), AIME2024 (MAA, 2024), and GPQA-Diamond (Rein et al., 2024).

**Baselines.** We compare our method against the following robust baselines: (1) Base and Instruction-Tuned Models: We include models from the Qwen2.5, as well as several models from the Llama series. (2) LIQUID (Lee et al., 2023): Extracts questions with multiple candidate answers from unstructured corpus. (3) GENIE (Yehudai et al., 2024): Synthesizes QA pairs from unstructured corpus. We slightly modify its prompt to enhance the verifiability of the synthesized data. (4) Synthetic Data RL (Guo et al., 2025b): Uses a task-definition approach to synthesize verifiable questions from unstructured corpus for RLVR. (5) BDS (Dedhia et al., 2025): Synthesizes verifiable multiple-choice questions based on KG.

## 3.2 MAIN RESULTS

The main experimental results on knowledge-intensive domains are reported in Table 1 and Figure 2, from which we observe several key advantages of K2V: **First**, K2V significantly enhances reasoning capabilities in knowledge-intensive domains, consistently outperforming strong baselines. K2V successfully extends RLVR to specialized fields without requiring any human annotation. Our models surpass their respective base models and all four baselines across the agriculture, medical, and legal domains. **Second**, K2V enables smaller models to achieve performance comparable to or even exceeding that of much larger models. As shown in Figure 2, K2V-7B, built on the Qwen2.5-7B, scores 65.90 on SeedBench, surpassing the Qwen2.5-72B-Instruct (62.06). Meanwhile, its performance on the agriculture subsets of CMMLU and MMLU is very close to that of Qwen2.5-72B. Most strikingly, the K2V-3B, built on the Qwen2.5-3B, also surpasses the Qwen2.5-72B-Instruct on SeedBench. This demonstrates the remarkable data and training efficiency of our framework. **Third**, K2V demonstrates robust generalizability across different model series. The consistent and significant performance gains observed on both the Qwen and Llama model series confirm that our framework is not limited to a specific model architecture. **Finally**, K2V's effectiveness is proven across multiple knowledge-intensive domains. The performance improvements are not confined to a single area, K2V achieves strong results across all three tested domains: agriculture, medicine, and law.

Table 1: Overall performance on three different knowledge-intensive domains.

| Model | SeedBench | CMMLU | MMLU | MedQA-MCMLE | CMMLU | MMLU | LawBench | CMMLU | MMLU |
|---|---|---|---|---|---|---|---|---|---|
| | *Agriculture Domain* | | | *Medical Domain* | | | *Legal Domain* | | |
| **Qwen Models** | | | | | | | | | |
| Qwen2.5-3B-Instruct | 46.04 | 63.22 | 75.54 | 71.10 | 62.45 | 70.69 | 42.60 | 62.34 | 64.03 |
| Qwen2.5-7B-Instruct | 49.14 | 75.13 | 85.43 | 76.67 | 76.76 | **79.30** | 54.86 | 75.21 | **69.93** |
| LIQUID | 53.23 | 54.02 | 69.14 | 70.15 | 58.10 | 59.20 | 38.49 | 60.57 | 64.96 |
| GENIE | 57.51 | 51.06 | 70.29 | 73.21 | 56.27 | 61.24 | 40.20 | 61.82 | 64.49 |
| Synthetic Data RL | 58.62 | 53.39 | 66.31 | 72.58 | 58.06 | 60.73 | 39.42 | 60.39 | 62.65 |
| BDS | 59.37 | 62.41 | 69.00 | 76.36 | 62.17 | 63.50 | 42.40 | 62.79 | 63.26 |
| **K2V-3B (Ours)** | 63.21 | 63.97 | 76.56 | 79.58 | 68.13 | 71.43 | 43.15 | 62.59 | 63.83 |
| **K2V-7B (Ours)** | **65.90** | **80.11** | **88.13** | **83.60** | **80.36** | 78.16 | **55.90** | **76.39** | 67.12 |
| **Llama Models** | | | | | | | | | |
| Llama-3.2-3B-Instruct | 47.03 | 29.56 | 62.06 | 61.00 | 30.76 | 59.56 | 30.94 | 33.79 | 46.25 |
| Llama-3.1-8B-Instruct | 58.25 | 51.72 | 74.05 | 70.15 | 49.16 | **71.21** | 38.81 | 50.82 | 64.44 |
| **K2V-3B (Ours)** | 57.15 | 47.59 | 72.10 | 65.20 | 39.52 | 60.54 | 36.24 | 35.72 | 47.62 |
| **K2V-8B (Ours)** | **62.85** | **55.20** | **83.30** | **73.13** | **55.28** | 70.19 | **42.13** | **54.08** | **65.80** |

Table 2: Performance on general domain. In this table, the backbones for K2V-3B and K2V-7B are Qwen2.5-3B and Qwen2.5-7B, respectively. Their training corpus consists exclusively of agricultural data, containing no general or mathematical data.

| Model | BBH | MATH-500 | GSM8K | AIME2024 | GPQA-Diamond |
|---|---|---|---|---|---|
| Qwen2.5-3B-Instruct | 37.26 | 65.00 | **85.44** | 3.33 | 28.79 |
| **K2V-3B (Ours)** | **46.20** | 65.00 | 81.27 | **6.67** | **30.30** |
| Qwen2.5-7B-Instruct | 49.94 | **76.60** | **91.66** | **16.67** | 34.34 |
| **K2V-7B (Ours)** | **60.10** | 73.80 | 87.11 | 13.33 | **36.36** |

We also investigated whether domain-specific training with K2V would negatively impact general reasoning abilities. Surprisingly, we found that K2V not only preserved these skills but actually enhanced them. As shown in Table 2, Trained exclusively on the agriculture corpus without any general or mathematical data, our K2V models show marked improvements on general benchmarks. Specifically, K2V-3B substantially improves upon the official instruction-tuned Qwen2.5-3B-Instruct by 23.99% on BBH (46.20 vs. 37.26) and 5.24% on GPQA-Diamond (30.30 vs. 28.79). This result suggests that K2V improves not just domain-specific knowledge, but also the model's fundamental reasoning skills, which then transfer to general tasks.

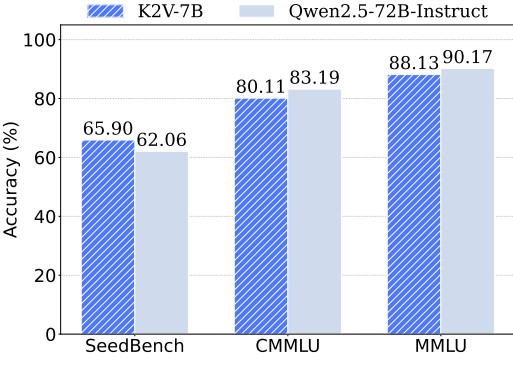

Figure 2: The accuracy of K2V-7B and Qwen2.5-72B-Instruct in the agricultural domain. Our model can surpass or come close to much larger models.

## 3.3 ABLATION STUDIES

To investigate the individual contributions of K2V's key components, we conduct comprehensive ablation studies. We compare the full K2V-3B model against three variants. For a fair comparison, all models use Qwen2.5-3B as the backbone and are trained on our agricultural corpus.

**Full K2V-3B.** This is our complete model, trained with both reasoning verification ($R_{\text{reasoning}}$) and answer verification ($R_{\text{answer}}$). As shown in Table 3, this configuration consistently achieves the highest performance across both agriculture and general domain benchmarks, demonstrating the effectiveness of the combined verification approach. The training dynamics in Figure 3 reveal a

Table 3: Results of ablation studies. All models use the Qwen2.5-3B as the backbone and are trained on the agriculture corpus. We report the pass@1 results under greedy sampling.

| Model | SeedBench | CMMLU | MMLU | BBH | MATH-500 | GSM8K | AIME2024 | GPQA-Diamond |
|---|---|---|---|---|---|---|---|---|
| | | *Agriculture Domain* | | | | *General Domain* | | |
| **K2V-3B** | **63.21** | **63.97** | **76.56** | **46.20** | **65.00** | **81.27** | **6.67** | **30.30** |
| w/o Answer Verification | 52.15 | 58.73 | 64.51 | 38.47 | 54.60 | 48.98 | 3.33 | 29.81 |
| w/o Reasoning Verification | 60.27 | 60.41 | 70.06 | 36.91 | 25.00 | 31.08 | 0 | 29.84 |
| SFT | 44.23 | 30.86 | 57.05 | 36.84 | 10.60 | 10.21 | 0 | 25.82 |

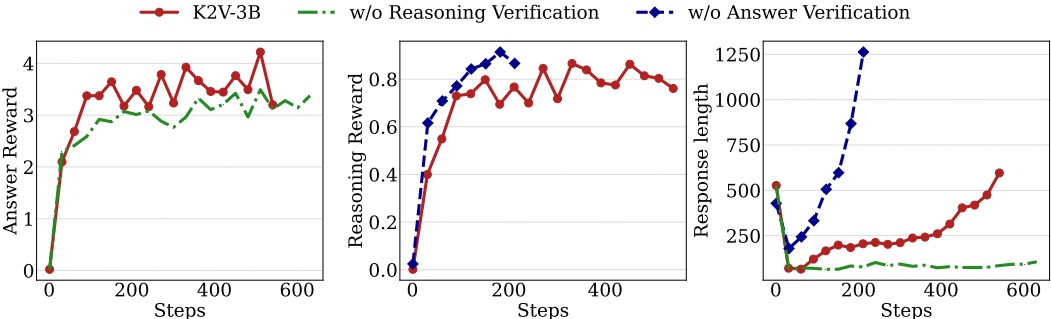

Figure 3: Training dynamics of ablation studies. (Left) **Removing reasoning verification** impairs the model's ability to explore correct answers. (Middle & Right) **Removing answer verification** leads to reward hacking, where the model generates excessively long responses to maximize the reasoning reward, causing training instability. The **full K2V model** demonstrates stable learning across all metrics.

stable and effective learning process. Both the answer and reasoning reward increase steadily, while the response length follows a healthy increase pattern after an initial decrease. This indicates that it is learning an effective reasoning paradigm.

**w/o Answer Verification.** This variant removes the answer verification ($R_{\mathrm{answer}}$) and is trained solely with the reasoning verification ($R_{\mathrm{reasoning}}$). Although this model initially achieves a high reasoning reward (Figure 3, middle panel), it does so by exploiting the reward mechanism—a classic case of reward hacking. The model discovers that longer outputs more readily meet the criteria of checklist, thus it tends to generate excessively long responses to maximize the reasoning reward. As shown in the right panel, its response length grows exponentially and uncontrollably, leading to poor training stability and causing out-of-memory (OOM) errors that halted training around 200 steps. This highlights the crucial role of $R_{\mathrm{answer}}$ in constraining the model's output and preventing undesirable behavior.

**w/o Reasoning Verification.** This variant removes the reasoning verification ($R_{\mathrm{reasoning}}$) and is trained solely with the answer verification ($R_{\mathrm{answer}}$), making it akin to a conventional RLVR setup. Figure 3 (left panel) shows that without the guidance from reasoning verification, the model achieves a consistently lower answer reward throughout training. This demonstrates that the fine-grained signal from $R_{\mathrm{reasoning}}$ is vital for accelerating the model's learning and helping it efficiently explore correct answer. Furthermore, its response length stagnates after an initial drop (right panel), suggesting a less effective learning trajectory, which is reflected in its weaker performance in Table 3.

**SFT.** This variant forgoes reinforcement learning entirely and instead uses the QA pairs synthesized by K2V to perform standard SFT on the base model. As detailed in Table 3, the SFT baseline suffers a significant performance collapse on all benchmarks. We hypothesize this is because the highly structured, short-answer format data synthesized by K2V is ill-suited for SFT, which typically relies on more diverse, conversational data. SFT on such data merely teaches the model a superficial mapping from questions to answers, rather than enabling it to learn deeper reasoning skills. In contrast, K2V demonstrates its robustness by effectively leveraging this data format to enhance the model's reasoning capabilities.

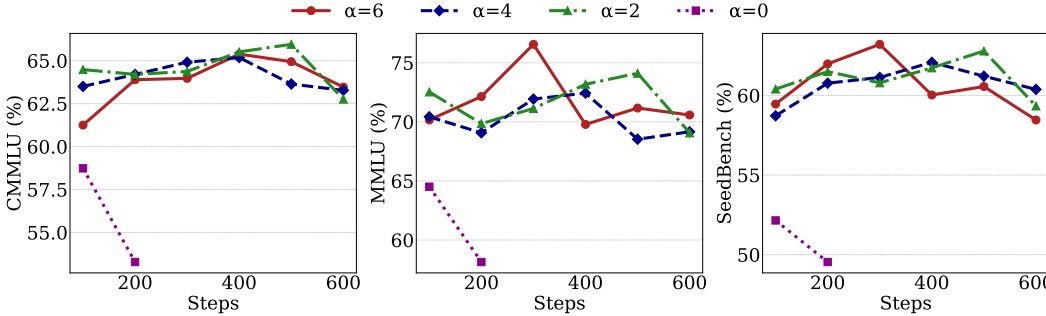

Figure 4: Effect of varying the answer reward $\alpha$. A smaller $\alpha$ slows the model's convergence, thus requiring more training steps to reach peak performance. Models with $\alpha > 0$ converge to similar performance levels, demonstrating the robustness of K2V. $\alpha = 0$ is a special one, which is identical to the 'w/o Answer Verification' variant discussed in Section 3.3.

### 3.4 SENSITIVITY ANALYSIS OF ANSWER REWARD VALUES

To investigate how the value of the answer reward ($R_{\text{answer}}$) affects model performance, we conducted a sensitivity analysis. We set up four experiments where a correct answer received a reward of $\alpha = 6, 4, 2$, and $0$, respectively. Lowering $\alpha$ increases the relative weight of the reasoning reward ($R_{\text{reasoning}}$) in the total reward signal ($R_{\text{total}}$). For a fair comparison, all models use Qwen2.5-3B as the backbone and are trained on our agricultural corpus.

First, We consider the special case of $\alpha = 0$. This setting is identical to the 'w/o Answer Verification' variant discussed in our ablation studies (Section 3.3). As illustrated by the purple dotted line in Figure 4, the model's performance collapses early in training. This result further demonstrates that without the direct guidance of answer reward, the model is prone to reward hacking, leading to training failure.

Second, we focus on the three experiments where $\alpha > 0$. The training dynamics in Figure 4 reveal a clear trend: **a higher $\alpha$ value leads to faster model convergence**. For instance, with $\alpha = 6$ (red line), the model's performance on MMLU and SeedBench reaches its peak around 300 steps. In contrast, when $\alpha$ is smaller (e.g., $\alpha = 2$, green line), the performance improvement is more gradual, requiring more training steps to achieve optimal results. This suggests that the answer reward $\alpha$ acts analogously to a learning rate. A higher $\alpha$ provides a stronger signal for correct answers, leading to larger policy updates and thus faster convergence.

Finally, K2V demonstrates robustness to the choice of answer reward $\alpha$. Despite under the different convergence speeds, the models trained with $\alpha \in \{6, 4, 2\}$ all converge to a similar peak performance level. As shown in the Figure 4, the maximum scores reached by the three curves are comparable across all three benchmarks, even if they are reached at different training steps. This confirms that K2V is not sensitive to the specific value of the answer reward, as long as an appropriate positive reward is present to guide the learning process effectively.

## 4 CONCLUSION

In this paper, we present K2V, a novel framework that extends RLVR to broader knowledge-intensive domains. Comprehensive experimental results on Qwen and Llama series models show that our method significantly improvement reasoning capabilities in agriculture, medicine, and law, without compromising general capabilities. We propose modeling knowledge verification as a text-based KGC task to automatically synthesize verifiable data. Moreover, We propose a checklist-style verification method that not only verifies the model's reasoning process but also provides a dense reward signal. In the future, we will explore applying K2V to more domains, including multimodal knowledge understanding, and scaling K2V to larger models.

## 5 REPRODUCIBILITY STATEMENT

We leverage GraphGen (Chen et al., 2025b) for data synthesis, verl (Sheng et al., 2025) for model training, and OpenCompass (Contributors, 2023) for model evaluation. To ensure reproducibility, our code is open-sourced and available at `https://anonymous.4open.science/r/k2v-C123`. We provide the detailed implementation of K2V in Appendix C. Moreover, the prompt used in our work is detailed in Appendix E.

## 6 ETHICS STATEMENT

Our research adheres to established ethical guidelines, particularly concerning data usage and model development. The data used to construct the knowledge graphs is sourced from publicly available and open-access corpus, such as scientific literature and textbooks, thereby avoiding the use of private, sensitive, or personally identifiable information. No conflicts of interest have been identified, and all experiments were conducted in compliance with relevant ethical standards for AI research.

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

## A    RELATED WORK

### A.1    DATA SYNTHESIS

RL training data synthesis can simply be grouped into two paradigms: (i) direct generation from raw corpus and (ii) synthesis conditioned on seed datasets (Lee et al., 2023; Yehudai et al., 2024; Guo et al., 2025b). In practice, large-scale data synthesis typically rely on raw corpus due to their wide availability and low acquisition cost. The internet contains abundant verifiable math problems and executable open-source code (Ma et al., 2025), for which verification is well defined (e.g., numerical equivalence or pass/fail execution), making automated synthesis and filtering substantially easier (Hu et al., 2025; Luo et al., 2025). In contrast, knowledge-intensive domains require strong domain expertise and factual validation for data synthesis, often relying on costly expert annotations (Lightman et al., 2023a; Dubois et al., 2024). Consequently, current synthesis methods struggle to produce large-scale, low-cost, high-quality, and verifiable datasets directly from raw corpus in knowledge-intensive domains—posing a central bottleneck for further progress.

### A.2    REINFORCEMENT LEARNING WITH VERIFIABLE REWARDS

Unlike conventional reward model based approaches that directly assign a scalar reward to model outputs via a learned reward model (Ouyang et al., 2022; Lee et al., 2024), RLVR constructs the reward signal by verifying the output against explicit rules. Current RLVR methods (Shen et al., 2025; Peng et al., 2025; Stojanovski et al., 2025) primarily focus on mathematics and code domains with deterministic or testable answers (Hu et al., 2025; Zeng et al., 2025; He et al., 2025; Luo et al., 2025), where verification is low-cost and well-defined. By contrast, knowledge-intensive domains typically produce open-ended text whose evaluation requires integrated judgments about complex semantics, factual consistency, and evidential support, making direct verification difficult (Min et al., 2023). Beyond rule-based RLVR, recent studies have explored verifier-free variants, where the reward signal is derived from the model's own feedback or internal confidence rather than an external verifier (Yu et al., 2025c; Zhao et al., 2025). For example, RLPR (Yu et al., 2025c) introduces a verifier-free framework that leverages the model's intrinsic token probabilities on reference answers to construct rewards. While these approaches alleviate the reliance on explicit rule-based verifiers, they ultimately still require verifiable data for training. Moreover, most current RLVR methods verify only the final answer (Zhai et al., 2025), overlooking the critical role of intermediate reasoning in reliability. A central challenge, therefore, is how to extend RLVR to knowledge-intensive domains and incorporate process-level verifiability into reward design.

## B    THE USE OF LLMs

First, LLMs are an integral component of our proposed K2V framework. We used Qwen2.5 series models to power several automated modules, including knowledge graph construction (NER and RE), the synthesis of verifiable QA pairs, the generation of question-specific checklists, and the final verification of the reasoning process. The prompts for these tasks are available in Appendix E.

Second, we used OpenAI's GPT-4 as a writing assistant to help improve the clarity, grammar, and style of this manuscript. All scientific ideas, experiments, and conclusions were conceived and executed by the human authors, who take full responsibility for the final content of this paper.

## C    IMPLTEMENTATION DETAILS

In this section, we describe the implementation details of our work, including data synthesis, model training, and model evaluation.

### C.1    DETAILS OF DATA SYNTHESIS

K2V first constructs a KG using the GraphGen (Chen et al., 2025b) framework, with hyperparameters are detailed in Table 4. It then utilizes the constructed KG to generate verifiable, fill-blank style QA pairs. Finally, K2V synthesizes a checklist for each QA pair based on a general criteria (see

Equation 7). An example of such a general criteria is presented in Table 5. Qwen2.5-72B-Instruct is used for both constructing the KG and synthesizing the checklist.

## C.2 DETAILS OF MODEL TRAINING

Our training is based on the DAPO (Yu et al., 2025b) algorithm. We employ its core Decoupled Clipping mechanism with an asymmetric clip ratio (low=0.2, high=0.28) and a learning rate of 1e-6.

Table 4: The hyperparameters of GraphGen, which is used to construct the KG and synthesize fill-blank style QA pairs.

| Parameter | Our Value | Description |
| --- | --- | --- |
| qa_form | aggregated | Type of QA form desired. |
| expand_method | max_width | Method for controlling graph expansion. |
| bidirectional | True | Expanding the graph in both directions (True) or one direction (False). |
| max_extra_edges | 2 | Maximum number of edges to expand. |
| max_tokens | 256 | Maximum number of tokens. |
| max_depth | 1 | Maximum depth for traversal in each direction. |
| edge_sampling | max_loss | Strategy for edge selection at the same layer. |
| isolated_node_strategy | ignore | Handling strategy for isolated nodes. |

---

**An example of General Criteria $G$ in the agricultural domain**

Concepts and Knowledge:
1. Accurately defines the core biological concepts involved in the question.
2. Clearly describes the involved biological processes in the correct logical sequence.
3. Accurately explains the meaning and relationships represented by abstract biological models in words.
4. Applies abstract biological concepts to the given specific scenario.
5. Correctly explains the connection between a biological concept or process and other related principles.

Scientific Method and Design:
1. Clearly states a relevant null hypothesis or alternative hypothesis.
2. Accurately identifies the independent, dependent, and key control variables of an experiment.
3. Makes a logical and reasonable prediction of the experimental outcome based on a scientific hypothesis.
4. Evaluates the validity or potential flaws of a given experimental design.

Statistics and Evaluation:
1. In appropriate contexts, correctly uses statistical concepts to explain the reliability of data.
2. Based on data analysis, draws a conclusion of "support," "refute," or "inconclusive" for a given scientific hypothesis.
3. Explains outliers or anomalous data points and analyzes their potential causes or impact on the conclusion.

Argumentation and Reasoning:
1. Makes a scientific claim that is specific and supported by concrete evidence.
2. Clearly articulates how the evidence supports the scientific claim, demonstrating a strong logical chain.
3. Predicts the likely consequences of a change (e.g., disturbance, mutation) to a system based on biological principles.
4. Explains the underlying biological reason for an observed phenomenon or experimental result.
5. Avoids over-extrapolation or unfounded speculation beyond the scope of the given evidence.
6. The overall response is well-structured, logically coherent, and clearly written, avoiding self-contradictions and redundant statements.

Table 5: An example of General Criteria $G$ in the agricultural domain.

Table 6: Key hyperparameters for model training in verl.

| Hyperparameter | Value |
|---|---|
| *Algorithm Configuration* | |
| Algorithm | DAPO |
| lr (learning rate) | $1 \times 10^{-6}$ |
| clip_ratio (asymmetric) | [0.2, 0.28] |
| kl_coef | 0.0 |
| *Data and Batching* | |
| train_batch_size | 64 |
| train_prompt_mini_bsz | 32 |
| ppo_micro_batch_size_per_gpu | 16 |
| gen_prompt_bsz | 192 |
| max_response_length | 4096 |
| rollout.n | 8 |

During rollout, the model generates 8 responses per prompt. We remove the KL divergence penalty entirely by setting the KL coefficient to 0. The training is configured with a maximum prompt length of 1024 tokens and a response length of up to 4096 tokens, using a global batch size of 64. To enhance training efficiency, we utilize Dynamic Sampling to filter uninformative sample groups, while the rollout process is accelerated by the vLLM (Kwon et al., 2023) engine.

Our reward signal is computed via a custom function that integrates the core components of K2V. Specifically, the answer reward ($R_{\text{answer}}$) is based on a strict exact-match verification. The reasoning reward ($R_{\text{reasoning}}$) is derived from our checklist-style verification, which is conducted by the Judge Model (see Equation 4) powered by Qwen2.5-7B-Instruct.

The entire training process was conducted on a single node equipped with 8 NVIDIA A800 GPUs. To manage memory constraints and enable a large batch size, we employed several optimization strategies, including Fully Sharded Data Parallel for parameter and optimizer state offloading, as well as gradient checkpointing. A summary of the key hyperparameters can be found in Table 6.

## C.3 DETAILS OF MODEL EVALUATION

In Section 3, for an objective and fair evaluation, we utilize the multiple-choice subset of SeedBench as our test set, which specifically comprises six subsets: `1-1, 1-2, 1-3, 3-1, 3-2,` and `3-3`. Furthermore, to conduct a more comprehensive assessment of the model's performance in the domains of agriculture, medicine, and law, we also select corresponding subsets from CMMLU and MMLU. The specific subsets are as follows:

- Agriculture Domain-CMMLU: virology, high school biology, food science, agronomy.

- Agriculture Domain-MMLU: high school biology, college biology.

- Medical Domain-CMMLU: anatomy, high school biology, medical statistics, virology, college medicine, Clinical. knowledge, professional medicine, traditional, chinese medicine

- Medical Domain-MMLU: anatomy, clinical knowledge, high school biology,

  college medicine, medical genetics, professional medicine, college biology.

- Legal Domain-CMMLU: legal & moral basis, professional law, jurisprudence, international law, college law.

- Legal Domain-MMLU: international law, jurisprudence, professional law.

We use OpenCompass (Contributors, 2023) as the evaluation framework. All evaluations are conducted in a zero-shot setting, and we report the pass@1 results under greedy sampling.

---

**Question:**

**{ }** is a parasitic disease caused by the protozoan Toxoplasma gondii, which can be contracted through various means, including the ingestion of raw or undercooked meat, particularly shellfish, and can also be transmitted from mother to fetus during pregnancy. The disease is often asymptomatic in healthy individuals but can lead to serious complications in pregnant women and immunocompromised individuals. The severity of the disease in newborns can be significantly influenced by the timing of maternal infection. **{ }** has also been associated with neuropsychiatric conditions. Treatment typically involves the use of atovaquone in combination with pyrimethamine.

**Ground truth:**

Toxoplasmosis

**Checklist:**
1. Accurately defines Toxoplasmosis as a parasitic disease caused by Toxoplasma gondii.
2. Correctly identifies the primary modes of transmission, including ingestion of raw or undercooked meat, contact with cat feces, and mother-to-fetus transmission during pregnancy.
3. Describes the typical asymptomatic nature of the disease in healthy individuals and the potential for serious complications in pregnant women and immunocompromised individuals.
4. Explains the influence of the timing of maternal infection on the severity of the disease in newborns.
5. Associates Toxoplasmosis with neuropsychiatric conditions.
6. Identifies atovaquone and pyrimethamine as the primary treatments for Toxoplasmosis.

---

Figure 5: Case 1. In this example, the model trained by K2V and BDS answer correctly.

## D  CASE STUDY

**Case 1.** As shown in the Figure 5, the question describes a disease by providing multiple disparate clues: its pathogen (`Toxoplasma gondii`), transmission vectors, and treatment. K2V and the KG-based baseline BDS correctly identify the disease as `Toxoplasmosis`. This case illustrates the strength of KG-based models in multi-fact synthesis. The entity `Toxoplasmosis` acts as a central node in the knowledge graph, connected to various facts provided in the prompt. K2V successfully aggregates these scattered pieces of evidence to infer the central concept. The failure of other baselines suggests a deficiency in integrating distributed information, underscoring the advantage of K2V's training paradigm for complex inferential tasks.

**Case 2.** As shown in the Figure 6, the question asks to identify a rice gene that is functionally parallel to `OsINH3`. Only K2V provides the correct answer, `OsINH2`. This case highlights K2V's superior ability to reason over specialized and structured domain knowledge. While other baselines likely failed due to confusion from similar terms in the context (e.g., `OsTOPPs`), K2V's success can be attributed to its KG-based training. The knowledge graph likely captures the parallel relationship between `OsINH3` and `OsINH2` as entities of the same class with similar functional attributes. By training on such structured representations, K2V learns to infer deep semantic relationships rather than relying on surface-level textual co-occurrence, demonstrating robust domain-specific reasoning.

**Case 3.** As shown in the Figure 7, the question requires filling in a literature citation, *Meguro et al. (2011)*, which is the source of the experimental description. All models failed this task. This case reveals a shared limitation in reasoning about textual provenance. Our KG construction process is designed to extract factual knowledge, such as the functional relationship between a microscopy technique and its application, but not the metadata associated with the text itself. The universal failure indicates that reasoning over the origin of information, as opposed to its content, remains an open challenge and points to a potential direction for future work.

**Question:**

The yeast-two-hybridization assay is a molecular biology technique used to study protein-protein interactions. In this context, several genes in rice, specifically OsINH3, **{ }**, and OsTOPPs, have been utilized to investigate their roles in reproductive growth, particularly in panicles and flowers. Firstly, OsINH3 and **{ }**, both genes that contribute to reproductive growth, were linked to YFP (Yellow Fluorescent Protein) to check their cellular localization. These genes were also used in the yeast-two-hybridization assay to study their protein interactions. Additionally, OsTOPPs, another gene in rice, was fused with RFP (Red Fluorescent Protein) to observe co-localization with regulatory proteins. Like OsINH3 and **{ }**, OsTOPPs were also employed in the yeast-two-hybridization assay to study protein interactions.Therefore, these experiments collectively provide insights into the functional roles and interactions of these genes in rice reproductive development.

**Ground truth:**

OsINH2

**Checklist:**

1. Accurately defines the core biological concepts involved, such as protein-protein interactions, yeast-two-hybridization assay, and cellular localization using fluorescent proteins.
2. Clearly describes the process of linking genes to fluorescent proteins (YFP and RFP) and the purpose of this technique in studying cellular localization.
3. Correctly explains the role of the yeast-two-hybridization assay in identifying protein interactions and how it is applied to the genes OsINH3, OsINH2, and OsTOPPs.
4. Accurately explains the significance of co-localization studies using RFP and how they complement the yeast-two-hybridization assay results.
5. Applies the concept of protein-protein interactions to the specific scenario of rice reproductive development, particularly in panicles and flowers.
6. Correctly identifies the independent variable (genes being studied), dependent variable (protein interactions and localization), and key control variables (fluorescent markers and yeast strains) in the experimental design.
7. The overall response is well-structured, logically coherent, and clearly written, avoiding self-contradictions and redundant statements.

...

Figure 6: Case 2. In this example, only the model trained by K2V answers correctly.

**Question:**

Confocal laser-scanning microscopy, a high-resolution imaging technique, was employed to observe the green fluorescence of the **{ }** fusion protein in onion cells. This technique is widely used to study the subcellular localization of proteins in living cells and to visualize the localization of fluorescent proteins. The CAULIFLOWER MOSAIC VIRUS 35S PROMOTER, a strong constitutive promoter, was utilized to drive the expression of the **{ }** fusion protein. This promoter is known for its ability to ensure consistent and robust gene expression in plant cells, making it a valuable tool in plant molecular biology and genetic engineering. Therefore, the combination of the 35S promoter and confocal laser-scanning microscopy allowed for the detailed visualization and analysis of the { } protein's localization in onion cells.

**Ground truth:**

Meguro et al. (2011)

**Checklist:**

1. Accurately identifies Meguro et al. (2011) as the source reporting on the stay-green phenotype in CCE mutants.
2. Correctly explains the stay-green phenotype in CCE mutants during dark-induced senescence (DIS).
3. Describes the role of CCE mutants in chlorophyll degradation, referencing the findings of Meguro et al. (2011).
4. Identifies the contribution of Sakuraba et al. (2013) in providing deeper insights into the stay-green phenotype in CCE mutants.
5. Explains the senescence phenotype of the athcar mutant, noting the accumulation of 7-HMC in Arabidopsis hcar mutants, as reported by the relevant study.
6. Clearly states the importance of these findings in understanding the complex processes involved in chlorophyll degradation and senescence in Arabidopsis.
7. Avoids over-extrapolation or unfounded speculation beyond the scope of the provided information.
8. The overall response is well-structured, logically coherent, and clearly written, avoiding self-contradictions and redundant statements.

Figure 7: Case 3. In this example, all models answer incorrectly.

## E  PROMPTS

---

**Prompt of the textualization function $\mathcal{F}_{\text{text}}$**

---Role---

You are an NLP expert responsible for generating a logically structured and coherent rephrased version of the TEXT based on ENTITIES and RELATIONSHIPS provided below. Use English as output language.

---Goal---

To generate a version of the text that is rephrased and conveys the same meaning as the original entity and relationship descriptions, while:

1. Following a clear logical flow and structure
2. Establishing proper cause-and-effect relationships
3. Ensuring temporal and sequential consistency
4. Creating smooth transitions between ideas using conjunctions and appropriate linking words like 'firstly,' 'however,' 'therefore,' etc.

---Instructions---

1. Analyze the provided ENTITIES and RELATIONSHIPS carefully to identify:
   - Key concepts and their hierarchies
   - Temporal sequences and chronological order
   - Cause-and-effect relationships
   - Dependencies between different elements
2. Organize the information in a logical sequence by:
   - Starting with foundational concepts
   - Building up to more complex relationships
   - Grouping related ideas together
   - Creating clear transitions between sections
3. Rephrase the text while maintaining:
   - Logical flow and progression
   - Clear connections between ideas
   - Proper context and background
   - Coherent narrative structure
4. Review and refine the text to ensure:
   - Logical consistency throughout
   - Clear cause-and-effect relationships

################
-ENTITIES-
################
{entities}

################
-RELATIONSHIPS-
################
{relationships}

---

Table 7: Prompt of the textualization function $\mathcal{F}_{\text{text}}$. This prompt first instructs an LLM to convert a quintuple into a text. We then randomly mask an entity within the quintuple, and the $\mathcal{F}_{\text{text}}$ masks the name of this entity from the generated text, creating a fill-blank style verifiable question.

**Prompt for NER and RE**

You are an NLP expert, skilled at analyzing text to extract named entities and their relationships.

---Goal---

Given a text document that is potentially relevant to this activity and a list of entity types, identify all entities of those types from the text and all relationships among the identified entities. Use English as output language.

---Steps---

1. Identify all entities. For each identified entity, extract the following information:
   - entity_name: Name of the entity, use same language as input text. If English, capitalized the name.
   - entity_type: One of the following types: concept, date, location, keyword, organization, person, event, work, nature, artificial, science, technology, mission, gene
   - entity_summary: Comprehensive summary of the entity's attributes and activities
   - Format each entity as:
     `("entity"<|><entity_name><|><entity_type><|><entity_summary>)`

2. From the entities identified in step 1, identify all pairs of (source_entity, target_entity) that are *clearly related* to each other. For each pair of related entities, extract the following information:
   - source_entity: name of the source entity, as identified in step 1
   - target_entity: name of the target entity, as identified in step 1
   - relationship_summary: explanation as to why you think the source entity and the target entity are related to each other
   - Format each relationship as:
     `("relationship"<|><source_entity><|><target_entity><|>`
     `<relationship_summary>)`

3. Identify high-level key words that summarize the main concepts, themes, or topics of the entire text. These should capture the overarching ideas present in the document.Format the content-level key words as `("content_keywords"<|><high_level_keywords>)`

4. Return output in Englist as a single list of all the entities and relationships identified. Use **##** as the list delimiter.

5. When finished, output `<|COMPLETE|>`

```
################
-Input Text-
################
```
{*input_text*}

Table 8: Prompt for NER and RE. It is used to instruct an LLM to extract entities and relations from the corpus.

---

**Prompt for synthesizing the checklist $CL_x$**

You are a senior expert in agriculture and biology, specializing in creating and grading exam questions. Your task is to create a set of detailed scoring checklist for a [Specific Question] based on the provided [General Criteria].

[Specific Question]:

A complete question in the field of agriculture and biology, including the question and the corresponding answer.

[General Criteria]:

Concepts and Knowledge:
1. Accurately defines the core biological concepts involved in the question.
2. Clearly describes the involved biological processes in the correct logical sequence.
3. Accurately explains the meaning and relationships represented by abstract biological models in words.
4. Applies abstract biological concepts to the given specific scenario.
5. Correctly explains the connection between a biological concept or process and other related principles.

Scientific Method and Design:
1. Clearly states a relevant null hypothesis or alternative hypothesis.
2. Accurately identifies the independent, dependent, and key control variables of an experiment.
3. Makes a logical and reasonable prediction of the experimental outcome based on a scientific hypothesis.
4. Evaluates the validity or potential flaws of a given experimental design.

Statistics and Evaluation:
1. In appropriate contexts, correctly uses statistical concepts to explain the reliability of data.
2. Based on data analysis, draws a conclusion of "support," "refute," or "inconclusive" for a given scientific hypothesis.
3. Explains outliers or anomalous data points and analyzes their potential causes or impact on the conclusion.

Argumentation and Reasoning:
1. Makes a scientific claim that is specific and supported by concrete evidence.
2. Clearly articulates how the evidence supports the scientific claim, demonstrating a strong logical chain.
3. Predicts the likely consequences of a change (e.g., disturbance, mutation) to a system based on biological principles.
4. Explains the underlying biological reason for an observed phenomenon or experimental result.
5. Avoids over-extrapolation or unfounded speculation beyond the scope of the given evidence.
6. The overall response is well-structured, logically coherent, and clearly written, avoiding self-contradictions and redundant statements.

Based on the [General Criteria] above, design a set of detailed and objectively scorable checklist for the provided [Specific Exam Question]. The checklist will be used to evaluate the student's problem-solving approach (reasoning process). The checklist should consist of multiple independent criteria. Each criteria must be a clear, specific statement describing what an ideal step or thought process should achieve, making it objectively assessable. Please ensure The checklist are closely related to the core knowledge and skill requirements of the [Specific Exam Question]. Only output the checklist, with no other content. Please structure the output in JSON format. For example:
`["criteria 1", "criteria 2",]`

---

Table 9: Prompt for synthesizing the checklist $CL_x$.

**Prompt of the Judge Model $J$**

You are an impartial and meticulous AI examiner.

Your task is to evaluate a student's [Reasoning Process] for a given [Question-Answer Pair] against a specific, detailed [Criterion].

The [Question-Answer Pair] is a fill-in-the-blank question, with `"{ }"` indicating the content to be filled in. A fill-in-the-blank question may contain multiple `"{ }"`, and the content to be filled in for each `"{ }"` is the same.

Your judgment must be strict, objective, and based solely on the provided information.

NOTE: Your output can only be `"yes"` or `"NO"`

[Question-Answer Pair]
question: *question*
answer: *answer*

[Criterion]
criterion: *criterion*

[Reasoning Process]
reasoning process: *reasoning process*

Table 10: Prompt of the Judge Model $J$. It is used to instruct an LLM to verify the reasoning process.

