# OpenReview forum: "Knowledge-to-Verification: Unlocking Reinforcement Learning with Verifiable Rewards for LLMs in Knowledge-Intensive Domains"
_ICLR.cc/2026/Conference — ICLR 2026 Conference Withdrawn Submission_

### Official Review · Reviewer_zqxm · 2025-10-28

**Soundness:** 2
**Presentation:** 3
**Contribution:** 3
**Rating:** 4
**Confidence:** 2

**Summary:**

This paper proposes **Knowledge-to-Verification (K2V)**, a framework that extends reinforcement learning with verifiable rewards (RLVR) to knowledge-intensive tasks such as those in medicine, agriculture, and law. K2V introduces two key innovations: (1) it synthesizes verifiable reward signals by converting domain-specific text into structured knowledge graphs and masking entities to generate fill-in-the-blank questions (quintuple completion); and (2) it defines reasoning verification via checklist-based criteria that evaluate not just final answers but intermediate reasoning steps. Experiments across three domains and two model families (Qwen and LLaMA) demonstrate strong performance gains on both domain-specific and general reasoning benchmarks.

This work takes an important step toward scaling RLVR beyond formal domains into open-domain knowledge reasoning. The engineering is clean and the empirical results are strong. However, the **lack of justification for task design**, **overreliance on checklist assumptions**, **absence of mechanism analysis for generalization**, **reliance on structured knowledge availability**, and **heavy dependence on LLM-generated components throughout the pipeline** limit its scientific contribution and practical applicability. With deeper theoretical insight or empirical analysis into how and why the system works, and clearer evidence of generalization to unstructured domains, this could become a more foundational paper.

**Strengths:**

* **Scalable reward generation**: K2V demonstrates that it is possible to generate verifiable, dense supervision without human annotation, enabling RLVR to scale to previously inaccessible domains.

* **Strong empirical performance**: The framework improves task accuracy across multiple knowledge-rich domains and surprisingly also improves general reasoning benchmarks like BBH and GPQA-Diamond, even when training only on domain-specific data.

* **Verified reasoning process**: Employs a novel checklist-style verification method that validates intermediate reasoning steps, providing dense reward signals and mitigating flawed reasoning.

* **Well-structured pipeline**: K2V offers a complete, modular framework with well-defined training signals, leveraging knowledge graph construction and rubric-based verification.

* **Informative ablations**: The reward hacking analysis (Fig. 5) and training dynamics clearly show how omitting either answer verification or reasoning verification weakens the model.

* **Clear writing and reproducibility**: The paper is well-written and supported by comprehensive appendices with implementation details and case studies.

**Weaknesses:**

### 1. **Heavy Reliance on LLM-Generated Components to Construct Training Data**

The entire K2V pipeline critically depends on LLM-generated outputs at multiple stages, creating a **quality bottleneck** that is not adequately addressed. In Sec. 2.1 (lines 142-186), the textualization function $\mathcal{F}_{\text{text}}$ uses a generative LLM to convert masked quintuples into fill-in-the-blank questions (Equation 2, line 165; Table 7), while Section 2.2 (lines 187-237) relies on LLMs both to synthesize the question-specific checklist (Equation 7, line 236; Table 9) and to serve as the judge model $J$ for verifying reasoning processes (Equation 4, line 210; Table 10). This creates a **circular dependency** where LLMs generate training data and verification criteria for training other LLMs, potentially amplifying existing flaws rather than correcting them. The quality of training data and verification is fundamentally bounded by the capabilities of the generation LLM, and **error propagation** occurs when errors in any component (incorrect question formulation, inappropriate checklist criteria, or inaccurate verification judgments) compound through the entire pipeline without independent validation. Furthermore, both checklist generation and the judge model rely on the same or similar LLMs (Qwen series), potentially leading to **lack of verification diversity** and overfitting to a particular model's reasoning style rather than learning general reasoning capabilities. The paper provides **no robustness analysis** of LLM-generated component quality, no ablation studies with different LLMs for generation/verification, no human validation of generated checklists, and no comparison with human-generated verification criteria.

### 2. **Verification Framework Assumptions**

The checklist-style verification assumes that reasoning can be decomposed into binary, independently verifiable steps (Sec. 2.3, line 258). However, this assumption may not hold in domains where reasoning is holistic, subjective, or integrative. Moreover, criteria are derived from AP Biology rubrics (line 230, footnote 1 at line 268; Appendix A.1; Table 5) and reused for medicine and law without domain adaptation or justification.

### 3. **Shallow Understanding of Learned Reasoning**

What analysis distinguishes genuine reasoning improvements from artifacts of training format or model scaling? The absence of transfer examples or generalization tests involving unseen entity types or reasoning chains makes it difficult to assess whether the approach truly enhances reasoning capabilities (Sec. 3.2, lines 310-323).

## Minor Issues

1. Why were experiments on LIQUID GENIE Synthetic Data RL conducted only on Qwen models and missing from Llama models? (Table 1, line 327)


2. **Limited Domain Evaluation and Generalization Analysis**: Empirical results are shown for only three domains (agriculture, medicine, legal) (Sec. 3.1, lines 277-287). The model's improvement on general reasoning benchmarks (e.g., BBH, GPQA-Diamond) is one of the most surprising and promising results (Table 1, lines 324-339; Table 2, lines 341-351), but **the mechanism remains unexplored**. It's unclear whether the gains arise from better reasoning, memorization of patterns, improved formatting, or regularization effects of dense reward.

**Questions:**

1. How robust is the K2V pipeline to the quality and diversity of LLM-generated components (Sec. 2.1-2.2, lines 142-237), and have the authors validated these components against human judgments or alternative generation models?

2. Given that the checklist-style verification was derived from AP Biology rubrics (line 230, Table 5), what evidence supports its generalization to medicine and law domains without domain-specific adaptation?

---

### Official Review · Reviewer_w6tV · 2025-10-29

**Soundness:** 3
**Presentation:** 3
**Contribution:** 3
**Rating:** 4
**Confidence:** 3

**Summary:**

The paper introduces K2V that extends Reinforcement learning with verifiable rewards (RLVR) to knowledge-intensive tasks by (i) building an LLM-derived knowledge graph (via GraphGen and Qwen2.5-72B) to synthesize fill-blank QA pairs, and (ii) generating QA specific checklists that a judge LLM (Qwen2.5-7B) uses to verify the model’s reasoning. The total RL reward is: format + binary answer match + checklist pass rate. This work conducts experiments across three domains (agriculture, medicine, and law) using the Qwen and Llama model series. The result shows that K2V has improved compared to baselines and most synthetic data generation methods. Ablations show both answer verification and checklist-based process rewards are necessary.

**Strengths:**

1. The primary strength of this paper is the proposed K2V framework, which generates the QA pairs and checklist from raw unstructured text without manual annotation and judges them by an LLM without human supervision.

2. The empirical results demonstrate that the K2V-trained models (K2V-7B) even outperform larger instruction-tuned baseline models (Qwen2.5-72B-Instruct) on the SeedBench benchmark in Figure 2. Table 1 shows that K2V exhibits improvement on most baseline models and synthetic data generation methods.

3. The authors examine the Qwen and Llama families and the results indicate the K2V’s generalization. The ablation studies (Table 3, Figure 3) convincingly demonstrate the necessity of each component of K2V.

**Weaknesses:**

1. The proposed K2V significantly relies on LLM as the judge, but there is no evaluation on the accuracy of LLM judgment. (1) First, the generated QA pair and checklist will include noise, where the failure case in Figure 7 is one example. But the paper does not investigate the rate of simple factual errors in its KG. (2) Second, the LLM judge’s pass/fail decisions are not validated. A human expert audit of a sample dataset is needed to validate the precision of the KG and the accuracy of the judge.

2. Is there any knowledge overlap between the benchmark and training data, especially in the agriculture domain? The model's performance may be due to memorization, not improved reasoning, and a rigorous contamination check is missing.

3. The data synthesis method creates "fill-blank style" questions by masking one entity from a 2-hop (quintuple) path in the KG and generates a checklist based on QA pair. This is a very simplistic and highly synthetic task format. The claim of improving "complex reasoning" is overstated. In Figure 5, several checklist items look like topical fill-in cues rather than items for reasoning verification.

4. In Table 1 (legal MMLU), some w/o reasoning verification synthetic-data methods (e.g., LIQUID, GENIE) outperform K2V-3B. To judge scaling fairly, please also report 7B results for those baselines.

5. K2V uses Llama Instruct backbones, while Qwen runs start from Base. Why not choose the llama base model as a backbone?

6. Table 1 would read more cleanly if you distinguished families, e.g., K2V-Qwen-3B vs K2V-Llama-3B, instead of the generic “K2V-3B.”

**Questions:**

Please refer to the weakness section.

---

### Official Review · Reviewer_Uvya · 2025-10-30

**Soundness:** 2
**Presentation:** 2
**Contribution:** 3
**Rating:** 4
**Confidence:** 4

**Summary:**

The paper proposes Knowledge-to-Verification (K2V), a framework that extends reinforcement learning with verifiable rewards (RLVR) to knowledge-intensive domains such as agriculture, medicine, and law. It first transforms domain corpora into knowledge graphs by using LLM-based extraction to identify entities and relations, forming quintuples. Each quintuple is then converted into a fill-in-the-blank QA pair by masking one entity, enabling automatic answer verification. K2V further introduces checklist-based reasoning verification, where a judge model evaluates intermediate reasoning steps using question-specific checklists to provide process reward signals. The combined reward, including format, correctness, and reasoning quality, guides RL training. Experiments show that K2V-trained 3B/7B/8B models outperform baseline models in domain-specific tasks and maintain certain general reasoning abilities. Ablations further indicate both answer and reasoning rewards are necessary for effective training.

**Strengths:**

- **Problem significance**: The paper aims to address an important and timely challenge: extending Reinforcement Learning with verifiable rewards (RLVR) to knowledge-intensive domains, where the answers are not easy to verify and the structured data for training is limited. It presents an end-to-end pipeline that converts unstructured text corpora into verifiable RL training signals, marking a meaningful step toward scalable knowledge reasoning.
- **Method novelty**: (i) The proposed quintuple-based textual KGC formulation that parses unlabeled text into a structured knowledge graph and generates verifiable problems is original and effective. (ii) The incorporation of checklist-based reasoning feedback introduces process-level evaluation into RLVR, complementing prior outcome-focused approaches.
- **Comprehensive experimentation**: The authors perform extensive experiments across three specialized domains and a general reasoning setting, training multiple models and baselines. The thorough comparisons and ablations provide a comprehensive view of the method’s effectiveness and robustness.

**Weaknesses:**

- **[Method] Unvalidated checklist quality**: The checklists are generated based on general principles and QA pairs but never evaluated for validity or alignment with human judgment. It remains unclear whether the checklist items represent desirable and necessary criteria. A human evaluation or consistency study would strengthen this component.
- **[Experiment] Lack of data leakage control**: The training corpora may overlap with evaluation benchmarks (e.g., SeedBench, CMMLU, MMLU), yet the paper does not report any contamination checks or KG-to-evaluation de-duplication. This raises concerns about potential data leakage that could inflate performance.
- **[Experiment] Unclear baseline comparability**: The implementation and training details for baseline methods (e.g., LIQUID, GENIE, Synthetic Data RL, BDS) are not clearly described, making it difficult to assess the reliability of the comparisons. Missing information includes backbone models, dataset sizes, and training configurations. Without this, the claimed advantage over baselines is not fully convincing.
- **[Experiment] General-domain degradation**: In Table 2, the K2V-7B model underperforms the instruct baseline on 3 out of 5 general benchmarks, yet the paper claims “marked improvements.” The authors should clarify how these mixed results support that statement.
- **[Clarity] Incomplete presentation details**: Some experimental information, such as total training samples or steps, the exact judge model used, and the training RL algorithm, is missing from the main text. Likewise, baseline implementation and configuration details are not provided in the appendix.

**Questions:**

1. How many training samples are used for each model, and what backbones, datasets, and training configurations are applied for LIQUID, GENIE, Synthetic Data RL, and BDS baselines?
2. The term "dense reward" usually refers to step-level feedback [1–3], but here it is applied post hoc to aggregated checklist scores. Could the authors clarify or adjust the terminology to avoid confusion?
3. The claim on lines 467–468 warrants clarification: if higher reward values accelerate convergence, why not use values larger than 6?
4. Have the authors tried training a single multi-domain model covering agriculture, medicine, and law, and how does it compare to domain-specific training?

[1] Eschmann, Jonas. "Reward function design in reinforcement learning." *Reinforcement learning algorithms: Analysis and Applications* (2021): 25-33.

[2] Cui, Ganqu, et al. "Process reinforcement through implicit rewards." *arXiv preprint arXiv:2502.01456* (2025).

[3] Lightman, Hunter, et al. "Let's verify step by step." *The Twelfth International Conference on Learning Representations*. 2023.

---

### Official Review · Reviewer_axmN · 2025-11-02

**Soundness:** 2
**Presentation:** 2
**Contribution:** 2
**Rating:** 2
**Confidence:** 3

**Summary:**

This paper proposes a novel knowledge-to-verfication (K2V) framework that automatically construct QA pairs from knowledge-intensive corpus and assign fine-grained reasoning quality reward with LLMs generated rubrics. Experiments show that K2V achieves good performance on different base model settings.

**Strengths:**

1. The proposed method saves human label costs by automatically generate QAs and reward rubrics and rewards.
2. The proposed K2V method achieves significant performance improvement on benchmarks.

**Weaknesses:**

1. Robustness analysis: The proposed K2V methods rely on a reliable KG construction process. The KG quality and how to keep such construction robust is not discussed in the paper.
2. Baselines: System-level baselines are important to demonstrate the real effectiveness of the proposed methods. For example, does such knowledge-intensive RL beats RL on math and code corpus?
3. Evaluation setting: single run pass@1 evaluation is too sensitive for reliable comparison. Subsets of benchmarks are much more smaller than the full set and consequently make this problem more severe.

**Questions:**

See Weakness.

---

### Author Response · Authors · 2025-12-04

We thank all reviewers for their feedback. In response to these comments, we are currently conducting additional experiments, optimizing our evaluation strategies, and revising the paper. We sincerely appreciate the time and effort dedicated by all reviewers.

---

### Note · Authors · 2025-12-04

I have read and agree with the venue's withdrawal policy on behalf of myself and my co-authors.